# DetCLIP: Dictionary-Enriched Visual-Concept Paralleled Pre-training for Open-world Detection

**Lewei Yao**[1,2*], **Jianhua Han**[2*], **Youpeng Wen**[3], **Xiaodan Liang**[3], **Dan Xu**[1],
**Wei Zhang**[2], **Zhenguo Li**[2], **Chunjing Xu**[2], **Hang Xu**[2†]
[1]Hong Kong University of Science and Technology, [2]Huawei Noah's Ark Lab
[3]Shenzhen Campus of Sun Yat-Sen University

## Abstract

Open-world object detection, as a more general and challenging goal, aims to recognize and localize objects described by arbitrary category names. The recent work GLIP formulates this problem as a grounding problem by concatenating all category names of detection datasets into sentences, which leads to inefficient interaction between category names. This paper presents DetCLIP, a paralleled visual-concept pre-training method for open-world detection by resorting to knowledge enrichment from a designed concept dictionary. To achieve better learning efficiency, we propose a novel paralleled concept formulation that extracts concepts separately to better utilize heterogeneous datasets (i.e., detection, grounding, and image-text pairs) for training. We further design a concept dictionary (with descriptions) from various online sources and detection datasets to provide prior knowledge for each concept. By enriching the concepts with their descriptions, we explicitly build the relationships among various concepts to facilitate the open-domain learning. The proposed concept dictionary is further used to provide sufficient negative concepts for the construction of the word-region alignment loss and to complete labels for objects with missing descriptions in captions of image-text pair data. The proposed framework demonstrates strong zero-shot detection performances, e.g., on the LVIS dataset, our DetCLIP-T outperforms GLIP-T by 9.9% mAP and obtains a 13.5% improvement on rare categories compared to the fully-supervised model with the same backbone as ours.

## 1 Introduction

Most state-of-the-art object detection methods [37, 39, 3, 55] can only recognize and localize a predefined number of categories. Their detection performance greatly relies on sufficient training data for each category, which requires expensive and time-consuming human annotations, especially for the rare classes that can only be distinguished from the expert. Even though a great effort has been made, existing publicly available detection datasets only have a limited number of object categories, for instance, 80 for COCO [31], 365 for Object365 [43], and 1203 for LVIS [16]. However, versatile open-world object detection is still out of reach mainly for two reasons: a) Due to the long-tail problem [16], finding sufficient examples for rare categories is surprisingly challenging; b) Developing a larger detection dataset requires extremely costly and labor-intensive manual annotation.

On the other hand, image-text pair data are cheap and abundant on the Internet. Recent vision-language (VL) pre-training methods (e.g., CLIP [36], ALIGN [26]) utilize those data to extend their open-domain capacity and have shown good zero-shot ability on various downstream classification tasks. Intuitively, some works [15, 51, 48, 12] try to extend a two-stage detector to an open-world

---

*Equal contribution, †Corresponding author: xu.hang@huawei.com

36th Conference on Neural Information Processing Systems (NeurIPS 2022).

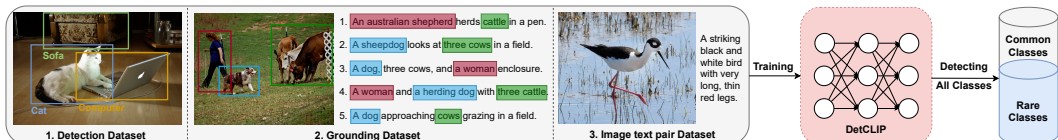

Figure 1: Task definition of open-world detection. In this paper, DetCLIP aims to develop a new pre-training pipeline with datasets from different domains (including detection dataset, grounding dataset and image-text pair dataset) as input, to deal with the open-world detection problem.

detector by distilling the learned image embeddings of the cropped proposal regions from a pre-trained VL model. However, this paradigm requires costly feature extraction from cropped images. There also exists discrepancy between instance-level features and conventional image-level features extracted from VL models. [14] proposes a self-training pipeline with pseudo labels (i.e., noun phrases in caption) generated by a pre-trained VL model [26] while the phrases in caption are always too limited to cover all the objects in an image. Moreover, their open-domain ability is determined by the performance of the pre-trained VL models.

Another line of work, GLIP [28] further proposes a grounding formulation of open-world detection by directly utilizing both detection and grounding data for pre-training. Specially for detection data, GLIP takes an image and a text prompt sentence that concatenates all the category names as input (i.e., sequential formulation). A text encoder will output features for this sentence and then GLIP aligns them with the region features extracted from the image encoder. However, restricted by the max length of the input token size of the text encoder, it is difficult for GLIP to operate on a large number of categories or extend to more detailed description of the categories. Furthermore, full attention matrix upon all the categories has to be learned in the text encoder, which is unnecessary and low-efficient for detection especially when the size of input categories increases.

To alleviate the above problems and further improve the open-domain ability, we present DetCLIP, a dictionary-enriched visual-concept paralleled pre-training method for open-world detection. Note that the "concepts" denotes the category names in detection data, and the phrases in grounding and image text pair data. Specifically, we first design *a novel paralleled concept formulation* to improve learning efficiency. Instead of feeding the whole prompt text sentence into the text encoder like GLIP, DetCLIP extracts each concept separately and parallelly feeds them into the text encoder (i.e., *paralleled formulation*). This paralleled formulation allows the model to avoid unnecessary interaction between uncorrelated categories/phrases, and produce a longer description for each concept. By converting detection data, grounding data, and image-text pair data into paralleled formulation, DetCLIP can be pre-trained under different types of supervision to support both localization and open-domain capability.

During pre-training, existing datasets often have a large domain gap and difference in their labeling space, e.g., the same object concept with different names or hierarchical/inclusive structures of different concepts. However, it is difficult to obtain these implicit relationships among concepts simply by their short names. In order to form a more unified concept space and provide prior knowledge (i.e., implicit relationships) for each input concept, we propose *a novel concept dictionary* to enrich our prompt text concepts during joint pre-training, as shown in Fig.2. Firstly, we construct the dictionary with concepts extracted from online resources and existing large-scale detection datasets

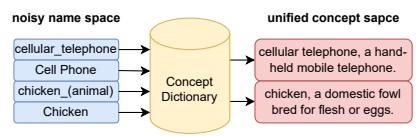

Figure 2: DetCLIP aims at unifying the noisy text name space into a joint concept space by a novel concept dictionary.

by considering both commonality and coherence. Based on this concept dictionary, DetCLIP can automatically enrich the current concept with the concepts and their descriptions existing in the dictionary to facilitate open-domain learning. To alleviate the partial label problem for the grounding and image-text pair data, DetCLIP further randomly samples concepts from the dictionary as negative samples to efficiently pre-train with the alignment loss. Besides, all concepts in the dictionary are served as the additional category inputs for label completion on the image-text pair data.

The proposed DetCLIP outperforms the state-of-the-art GLIP by large margins on large-scale open-world detection benchmark LVIS [16] with 1203 classes. Particularly, DetCLIP-T achieves around

9.9% mAP improvement over GLIP-T on LVIS without utilizing any images in LVIS during pre-training. Compared to the baseline ATSS [53] model (with the same swin-T [33] backbone) trained on LVIS, our DetCLIP-T achieves 2.3% mAP improvement in the total performance and 13.5% mAP improvement on the rare classes.

## 2 Related Work

**Vision-Language Pre-training (VLP)**. Current Vision-Language Pre-training is a natural extension and development of the successful pre-train-and-fine-tune scheme in the domains of natural language processing (NLP) [10, 1] and computer vision [11] community. Dual-stream methods such as CLIP [36] and ALIGN [26] have shown great zero-shot classification ability by performing cross-modal contrastive learning on large-scale image-text pairs from the Internet. Single-stream approaches [27, 21] directly model the interaction between the visual and textual embeddings together by a single transformer-based model, which can perform tasks such as image caption and VQA. Recent approaches such as VLMo [46] and BLIP [25] further explore a mixed architecture of single-stream and dual-stream models to enable a unified way of vision-language understanding and generation. However, those approaches usually focus on whole-image representation learning and the pre-trained models are usually designed for retrieval/generation tasks. Current vision-language pre-training approaches can not directly be applied to object detection task, i.e., a core computer vision task.

**General Object Detection.** Object detection is a core problem in computer vision. CNN-based object detection methods (one-stage detectors: YOLO [37], SSD [32] and ATSS [33] and two-stage detectors: Faster R-CNN [39] and R-FCN [7]) usually use a classifier to map ROI (Region Of Interest) features into categories. Further improvement of the CNN-based detectors such as K-Net [54] and Panoptic-FCN [29] further introduce dynamic kernels and replace the static kernels in the convolution layers to improve the flexibility of models. Recent transformer-based methods such as DETR [3] and Deformable DETR [55] try to formulate the object detection as a set prediction problem that can eliminate post-process NMS. Those methods are constrained to predefined categories, while our method tries to endow the detector with open-domain recognition ability which can detect any categories by learning a wide range of concepts.

**Zero-shot Object Detection/ Open-vocabulary Object Detection.** In the early setting of this area, zero-shot object detection aims to generalize the detector from known categories (training) to unknown categories (inference). Under this setting, various works [5] try to find relationships between existing categories and unknown categories through pre-trained semantic/text features [44, 35, 38, 4, 13] knowledge graphs [41, 19, 49, 47] and so on. However, the evaluation under this setting is not general enough since people usually simply split the class name into known/unknown categories in the single dataset and the transfer learning is still under similar domain. On the other hand, inspired by the success of vision-language~(VL) pre-training methods (e.g., CLIP [36]) and their good zero-shot ability, several methods attempt to perform zero-shot detection on a wider range of domains by leveraging a pre-trained VL model. For example, [15] tries to distill the learned image embeddings of the cropped proposal regions from CLIP [36]} to a student detector. [14] proposes a self-training pipeline which utilizes Grad-CAM [42] and ALBEF [26]. However, those methods are very slow because the feature extraction is repeated and the image-level representation may be sub-optimal for the instance-wise tasks. Recently, GLIP [28] and X-DETR [2] try to align region and language features using a dot-product operation and can be trained end-to-end on both grounding data and detection data. Our method aims to design an open-domain detector that learns new concepts efficiently and expands domain coverage from low-cost data from the Internet.

## 3 The Proposed Approach

This paper aims to develop a vision-language pre-training pipeline to enhance new concept learning for open-world detection. To achieve this goal, our DetCLIP leverages a new paralleled vision-concept per-training pipeline (Sec.3.1) for efficient training and a concept dictionary (Sec.3.2) to provide external knowledge to automatically enrich the current input concept and alleviate the partial-labeling problem of the grounding and image-text pair data. Our DetCLIP is pre-trained under hybrid supervision from detection data, grounding data and image-text pair data (Sec.3.3).

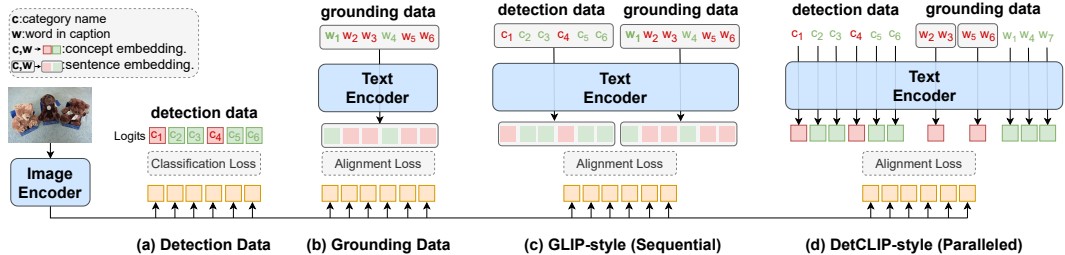

**Figure 3**: Comparison of different concept formulation strategies. (a) Traditional detection and (b) grounding training utilize different input formulations. (c) To utilize both detection and grounding datasets, GLIP converts the detection data into grounding format by formulating all classes into one sentence as input. (d) Our DetCLIP introduces a paralleled concept formulation to extract the phrase from the grounding data and regards each phrase as an individual input that is the same as the category name in the detection dataset. Red color denotes that the corresponding concept is in the image while the green indicates the concept is not in the image.

## 3.1 Paralleled Concept Formulation

A robust open-world detector is required to be trained with sufficient data covering enough vision concepts. Current available detection datasets lack enough concepts and have limited capacity due to the costly annotation. Leveraging data from other sources, e.g., grounding data and image-text pair data, is a feasible solution to augment the semantic coverage. To enable training with heterogeneous supervisions, we are required to find a unified formulation for different formats of data.

Fig.3 illustrates a comparison of the different concept formulation strategies that are used for pre-training with different types of data. As shown in Fig.3(a)-(b), traditional detection and grounding training utilize different input formulations. Detection treats each category as a fixed label and aligns region features to the pre-defined label space, while the grounding training leverages the whole caption sentence as input, models the attention between words, and adopts each token embedding in the output embeddings for alignment. To utilize both detection and grounding datasets for pre-training, GLIP [28] converts the detection data to phrase grounding data, by replacing the object classification logits with the word-region alignment scores, which are associated with a sentence that concatenates all the category names (see Fig.3(c)), i.e., the text input is ["person, bicycle, car, ... , toothbrush"].

We argue that this sequential form is not an effective formulation to model open-world object detection as a vision-language task because it (1) leads to unnecessary interaction between category names in the attention module; and (2) constraints the number of negative samples in contrastive learning due to the limited context length of text input. Ablation studies conducted for GLIP show that randomly shuffling the word order in the grounding training data can even bring a slight improvement for the downstream detection task, indicating that the noun phrase is more critical for the detection task, compared to the context information.

To address the above problem, our DetCLIP introduces a paralleled concept formulation to train with different data sources. As shown in Fig.3(d), we extract the concept noun phrase for each bounding box and then feed them into the text encoder individually to obtain the corresponding text embedding. In our paralleled design, context information is removed and model directly learns language features from each separate concepts, which is a more straightforward modeling for the detection task and improves the learning efficiency (see Fig.4). Furthermore, our paralleled design can enable easy expansion of the augment of category descriptions (see Sec.3.2.2). Detailed paralleled formulation for different types of data are as follows:

**Detection data:** Suppose there are $k$ positive categories in an image, we first pad the category number to $N$ by randomly sampling the negative categories, where $N$ is a

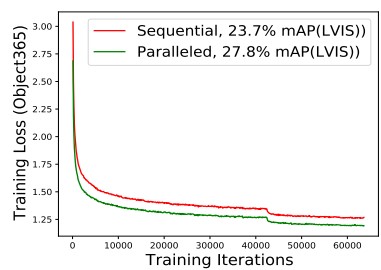

**Figure 4:** Efficiency comparison between two formulations. Paralleled formulation learns faster and achieves a better performance (27.8% v.s. 23.7%).

predefined number of the concepts for constructing the alignment loss. More specifically, the text input $P$ can be formulated as $\{p_n\}_{n=1}^N$, where $p_n$ represents for the $n$-th category name, e.g.,

$$P = [\text{"person", "bicycle", "car", ... , "toothbrush"}].$$

Then, we feed $N$ category names as separate sentences into the text encoder and use the embedding of [end of sentence] token as the text embedding for each category. The $N$ text embeddings are then concatenated and matched with ground truth bounding boxes to construct the alignment loss.

**Grounding data:** We extract positive phrases for bounding boxes (provided by the grounding annotations), and drop other words in the caption. To align the input format with detection data, we also pad the category number to $N$ by sampling negative categories from our proposed concept dictionary (see Sec.3.2.2). An example of input $P$ for grounding data can be:

$$P = [\text{"a woman", "a herding dog", "three cattle", "}neg_1\text{", ... , "}neg_m\text{"}],$$

where $neg_1$ to $neg_m$ are sampled negative categories from the constructed concept dictionary. Then similar steps with detection data can be conducted to construct the alignment loss.

**Image-text pair data:** Image-text pair data only contains images and the corresponding captions, while without any annotated bounding boxes. To obtain object-level dense labels, we first use a pre-trained class-agnostic region proposal network (RPN) to extract object proposals. Then a powerful pre-trained vision-language model like CLIP [36] or FILIP [50] is utilized to assign pseudo labels for the proposals (see Appendix for more details.). To tackle the concept-missing problem in the captions of image-text pair data, instead of using noun phrases in the captions as the candidate category names following [28, 14], we propose to assign open-domain categories to the object proposals via constructing a large-scale concept dictionary (see Sec.3.2.2). After obtaining object-level dense pseudo labels, we convert the image-text pair data to the same format as the detection and grounding data, and then a similar training procedure can be applied .

## 3.2 Concept Dictionary

In this work, we try to build an open-world object detector that can cover a wide range of concepts and can be applicable to different types of datasets. However, the existing detection/grounding/image-text-pair datasets have a very large domain gap and difference in their labeling space. For example, a boy can be annotated as "man", "child" or "people" in the different datasets. Moreover, there often exists hierarchical/inclusive relationships between different concepts. Knowing these implicit relationships can effectively facilitate the pre-training, however it is also clearly challenging to discover these relationships with only a limited set of concept names.

Therefore, we propose to build a large-scale concept dictionary to form a unified concept space for different data sources and explicitly provide the useful relationships between various concepts through definitions. For example, the car is defined as "a motor vehicle with four wheels usually propelled by an internal combustion engine", and the motorcycle is defined as "a motor vehicle with two wheels and a strong frame". By aggregating these definitions, we can conclude that the car and the motorcycle are both motor vehicles with a difference in the number of wheels.

### 3.2.1 Constructing the Concept Dictionary

To construct a unified concept dictionary $O$, we collect concepts from multiple sources: (1) noun phrases extracted from the large-scale image-text pair dataset, i.e., YFCC100m; (2) category names of existing public detection datasets (e.g., Objects365 [43], OpenImages [24]); (3) object names from the manually-collected concept database, i.e., Things [18]). For object names from the detection datasets and Things, we directly add them into the dictionary after deduplication. For noun phrases extracted from the YFCC, we filter out the concepts that appear less than a frequency of 100 or without definition in WordNet [34] to ensure the commonality. Our concept dictionary $O$ is then constructed by concatenating each word $o_l$ with its definition $def_l$ in WordNet: $O = \{o_l : def_l\}_{l=1}^L$, where $L$ is the number of concepts in our dictionary. The constructed dictionary covers about 14k concepts along with their definitions, and can be updated by directly adding new concepts and corresponding definitions. Based on the constructed concept dictionary, we then propose 2 techniques utilizing it to boost the pre-training in the following section.

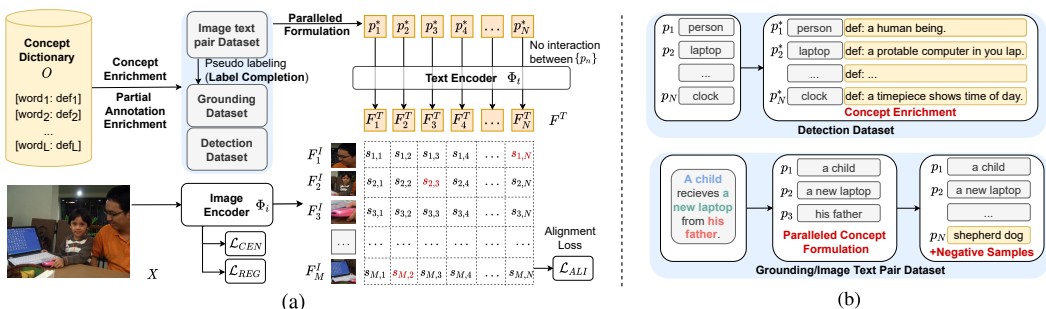

Figure 5: Overall architecture and the details of utilizing concept dictionary $O$. (a) DetCLIP contains an image encoder $\Phi_i$ to obtain region features $F^I$ and a text encoder $\Phi_t$ to get embeddings $F^T$ for each enriched concept $p_n^*$. Then the region-concept alignment loss $\mathcal{L}_{ALI}$ is performed. Note that the box regression loss $\mathcal{L}_{REG}$ is only adopted on detection datasets. (b) A concept dictionary $O$ is introduced to enrich the current concept with prior knowledge and provide negative category samples for construction of the alignment loss.

### 3.2.2 Knowledge Enrichment with Concept Dictionary

**Concept Enrichment.** Based on the designed concept dictionary, we first retrieve the definition for each input concept to provide the prior knowledge (see Fig.5(b)). During pre-training, for each concept $p_n$ in the training set, we can directly use its definition if $p_n$ is included in the dictionary $O$. If we cannot find a direct match in $O$, we will try to locate the most related concept in $O$ by calculating a similarity matrix $S' \in \mathbb{R}^L$. The $S'$ is calculated via the dot-product of the embeddings from a pre-trained text encoder such as FILIP [50] with $p_n$ and all concept names $\{o_l\}_{l=1}^L$ as input. Then we can find the most related concept $\{o_{l^*}, \text{where } l^* = argmax_l(S'(l))\}$ in the dictionary, to retrieve an approximate definition $def_{l^*}$. The $p_n$ is then enriched with the retrieved definition and reformatted as $\{p_n^*\} = \{p_n, def_{l^*}\}$. An example of the enriched text input $P^* = \{p_n^*\}_{n=1}^N$ is:

$$P^* = [\text{"person, a human being.", "bicycles, a wheeled vehicle that has two wheels and is moved by}$$
$$\text{foot pedals.", ... , "toothbrush, small brush has long handle used to clean teeth."]}$$

**Partial Annotation Enrichment.** In the grounding or image-text pair data, only main objects that people care about are labeled in the caption, which is known as the partial labeling problem. Compared with standard detection datasets which have sufficient positive and negative classes for each image, pre-training with grounding and image-text pair datasets encounters two severe issues: 1) lack of annotations of negative concepts for learning discriminative concept embeddings; 2) lack of annotations of partial positive concepts to efficiently train the model. For the first problem, DetCLIP randomly samples the concepts in the constructed dictionary $O$ as the negative concepts to construct the alignment loss, instead of directly padding empty inputs (Fig.5(b)). Note that since the number of concepts in the dictionary $O$ is large (i.e., about 14k), the probability that the sampled concepts are indeed in the image is extremely small. For the second problem, to perform label completion on image-text pair data during pseudo labeling, we add all the concepts in dictionary $\{o_l\}_{l=1}^L$ as the additional category inputs, instead of using the original noun phrase in the caption to calculate the similarity matrix. Therefore, the concepts shown in the image while not in the caption can also be labeled and then get pre-trained. An illustration is also shown in Fig.7 to qualitatively verify the effectiveness of label completion.

### 3.3 Model Architecture/Training Objective

As shown in Fig.5, the basic architecture of DetCLIP contains an image encoder $\Phi_i$ to generate the region features $F^I \in \mathbb{R}^{M \cdot D}$ from the input image $X$, and a text encoder $\Phi_t$ to obtain the text embeddings $F^T \in \mathbb{R}^{N \cdot D}$ for the concepts in $P^*$, where $M$, $N$ denote for the number of extracted regions and input concepts, respectively. Then the alignment loss is constructed by calculating the alignment score $S \in \mathbb{R}^{N \cdot M}$ for all region-text pairs.

$$F^I = \Phi_i(X), F^T = \Phi_t(P^*), S = \langle F^I, \text{Transpose}(F^T) \rangle \tag{1}$$

With the ground-truth alignment matrix $G \in \mathbb{R}^{N \cdot M}$, the whole training objective $\mathcal{L}$ can be written as:

Table 1: Zero-shot performance on LVIS [16] minival datasets. $AP_r$ / $AP_c$ / $AP_f$ indicate the AP values for rare, common, frequent categories, respectively. 'DH' and 'F' in GLIP [28] baselines stand for the dynamic head [8] and cross-modal fusion, respectively. Baselines with * are implemented with our code-base. GoldG+ denotes the GoldG plus the COCO [31] caption dataset.

| MODEL | BACKBONE | PRE-TRAIN DATA | LVIS | |
| --- | --- | --- | --- | --- |
| | | | AP | $AP_r$ / $AP_c$ / $AP_f$ |
| MASK-RCNN[17]* | SWIN-T | LVIS | 34.1 | 19.1 / 34.0 / 37.0 |
| ATSS[53]* | SWIN-T | LVIS | 33.6 | 19.7 / 32.4 / 37.2 |
| ATSS[53]* | SWIN-L | LVIS | 43.9 | 30.6 / 43.7 / 46.3 |
| MDETR[20] | RN101 | GOLDG+ | 24.2 | 20.9 / 24.3 / 24.2 |
| GLIP-T(A)[28] | SWIN-T+DH+F | O365 | 18.5 | 14.2 / 13.9 / 23.4 |
| GLIP-T(C)[28] | SWIN-T+DH+F | O365,GOLDG | 24.9 | 17.7 / 19.5 / 31.0 |
| GLIP-T[28] | SWIN-T+DH+F | O365,GOLDG,CAP4M | 26.0 | 20.8 / 21.4 / 31.0 |
| GLIP-L[28] | SWIN-L+DH+F | 4ODS,GOLDG,CAP24M | 37.3 | 28.2 / 34.3 / **41.5** |
| GLIPV2-T[52] | SWIN-T+DH+F | O365,GOLDG,CAP4M | 29.0 | - / - / - |
| DETCLIP-T(A) | SWIN-T | O365 | 28.8 | 26.0 / 28.0 / 30.0 |
| DETCLIP-T(B) | SWIN-T | O365, GOLDG | 34.4 | 26.9 / 33.9 / 36.3 |
| DETCLIP-T | SWIN-T | O365, GOLDG, YFCC1M | 35.9 | **33.2 / 35.7 / 36.4** |
| DETCLIP-L | SWIN-L | O365, GOLDG, YFCC1M | **38.6** | **36.0 / 38.3 / 39.3** |

$$\mathcal{L} = \mathcal{L}_{ALI}(S, G) + \alpha \cdot \mathcal{L}_{CEN} + \beta \cdot \mathcal{L}_{REG}, \qquad (2)$$

where $\mathcal{L}_{ALI}$, $\mathcal{L}_{CEN}$, and $\mathcal{L}_{REG}$ denote the alignment loss, the centerness loss, and the regression loss, respectively. $\alpha$ and $\beta$ represent the weight factor for $\mathcal{L}_{CEN}$ and $\mathcal{L}_{REG}$, respectively. Following GLIP [28], we adopt the ATSS [53] detector as our image encoder. We use the sigmoid focal loss [30] for $L_{ALI}$, the sigmoid loss for $L_{CEN}$, and the GIoU loss [40] for $L_{REG}$.

## 4 Experimental Results

**Implementation Details.** We pre-train all the models based on Swin-Transformer [33] backbones with 32 GPUs. AdamW optimizer [22] is adopted and batch size is set to 128. The learning rate is set to $2.8\text{x}10^{-4}$ for the parameters of the visual backbone and detection head, and $2.8\text{x}10^{-5}$ for the language backbone. Without otherwise specified, all models are trained with 12 epochs and the learning rate is decayed with a factor of 0.1 at the 8-th and the 11-th epoch. The max token length for each input sentence is set to 48. The number of the concepts $N$ in text input $P$ is set to 150 and the number of region features $M$ is determined by the feature map size and the number of pre-defined anchors. The loss weight factors $\alpha$ and $\beta$ are both set to 1.0. The training of DetCLIP-T with Swin-T backbone tasks 63 hours when 32 GPUs are used. MMDetection [6] code-base is used.

**Training Data.** Our model is trained with a hybrid supervision from different kinds of data, i.e., detection data, grounding data, and image-text pair data. More specifically, for detection data, we use a sampled Objects365 V2 [43] dataset (denoted as O365 in the following sections) with 0.66M training images. Here we do not use the whole dataset since training with sampled data is more efficient and suffices to demonstrate the effectiveness of our method. For grounding data, we use gold grounding data (denoted as GoldG) introduced by MDETR [20]. Moreover, following GLIP [28], we remove the training samples contained in LVIS [16] dataset for fair zero-transfer evaluation, which results in 0.77M training data. For image-text pair data, we perform object-level dense pseudo labeling on YFCC100m [45] dataset with a pre-trained CLIP [36] model, and sample a subset of 1M training images from the results containing objects with similarities scores above a given threshold. Finally, our training set contains a total of 2.43M images. Compared to GLIP's 27M training data, *DetCLIP only uses less than 10% data, but achieves better results*. More details are in Appendix.

**Benchmark Settings.** We evaluate our method mainly on LVIS [16] which contains 1203 categories. Following GLIP [28] and MDETR [20], we evaluate on the 5k minival subset and report the zero-shot *fixed* AP [9] for a fair comparison. We do not focus the performance on COCO [31] since it only contains 80 common categories that are fully covered by the training dataset Objects365 [43], which

may not sufficient to reflect generalization ability of a model in the open-domain detection setting. To further study the generalization ability of our method, following GLIP [28], we also evaluate the averaged AP on other 13 downstream detection datasets published on Roboflow[1].

## 4.1 Open-world Detection Results

We train our DetCLIP with two backbones, i.e., swin-T [33] and swin-L. Distinct from GLIP [28], which introduces additional heavy modules like Dynamic Head [8] and cross-modal fusion, we directly adopt the vanilla ATSS [53] as our vision encoder to keep our architecture as neat as possible. Following GLIP, for Swin-T backbone, we also train 3 versions of models, i.e., DetCLIP-T(A), DetCLIP-T(B) and a complete version DetCLIP-T, which differ in using different training data.

Table 1 reports the results on LVIS [16]. Results with LVIS as pre-training data stand for the fully-supervised models trained with annotated data. With our proposed paralleled formulation, introducing more training data from different sources can consistently improve the performance. I.e., comparing DetCLIP-T(A) trained with only detection data with DetCLIP-T trained with additional grounding and image-text pair data, we can observe a considerable performance gain (28.8% AP v.s. 35.9% AP). Besides, befitting from the proposed effective framework, our DetCLIP models outperform their GLIP counterparts by a large margin, i.e., DetCLIP-T(A) (resp. DetCLIP-T) surpasses GLIP-T(A) (resp. GLIP-T) by 10.3% (resp. 9.9%). Besides, DetCLIP-T also significantly outperforms GLIPv2-T by 6.9%. Note that our models are more lightweight (without heavy DyHead and cross-modal fusion) and trained with fewer epochs (our 12 vs. GLIP's 24) and much less data. In addition, our DetCLIP-T's zero-shot performance *even beats the fully-supervised model with the same backbone* by utilizing weak-annotated data like image-text pair data. Results on LVIS full validation dataset can refer to the Appendix.

**Efficiency Comparison.** To demonstrate the efficiency of our proposed DetCLIP, we directly compare the training and inference speed of DetCLIP-T with the GLIP-T [28] in Table 2. With the same setting of training with 32 V100 GPUs, the total training time for GLIP-T is about 10.7K GPU hours (5X than us) due to its heavy backbone and more image-text pair training data. On the other hand, DetCLIP-T achieves the 2.3 FPS (0.43 s/image) on a single V100 when performing inference on LVIS [16], while GLIP-T can only achieve 0.12 FPS (8.6 s/image). With much better training and inference efficiency, DetCLIP-T can still outperform GLIP-T 9.9% on LVIS.

Table 2: Efficiency comparison on LVIS.

| MODEL | TRAINING | INFERENCE |
|---|---|---|
| GLIP-T | 10.7K GPU HRS | 0.12 FPS |
| DETCLIP-T | **2.0K** GPU HRS | **2.3** FPS |

**Qualitative Visualizations.** We illustrate the bounding box predictions on LVIS [16] dataset from DetCLIP-T and GLIP-T [28] model in Fig.6. We can observe that by adopting the paralleled concept formulation and the external knowledge from the concept dictionary, our DetCLIP can outperform GLIP both on the accuracy and completeness of the predicted labels, especially on the rare classes.

## 4.2 Ablation Studies

Table 3 studies the effectiveness of two core components of DetCLIP, i.e., the paralleled formulation and the knowledge enrichment with concept dictionary. The first row stands for our implementation of a GLIP-A [28] model, which is modeled with sequential formulation and uses only detection data for training. Due to the implementation discrepancy, our version can achieve 23.7% zero-shot AP on LVIS [16], which is higher than the official's 18.5% and serves as a stronger baseline.

First, applying the paralleled formulation can bring significant improvements (row 2), boosting the performance to 27.8%. This indicates that paralleled formulation is much more effective than sequential formulation for modeling object detection as a vision-language task. However, directly applying the same approach to a larger scale dataset with heterogeneous supervisions, e.g., detection plus grounding, can hurt the performance (row 4). We speculate this is because paralleled formulation weakens the interaction between text concepts, resulting in the model's inability to effectively construct the connections between semantic-related concepts. Therefore, we introduce word definitions to

---

[1]`https://public.roboflow.com/object-detection`

Table 3: Ablation Studies of different components on LVIS [16] minival and other 13 detection datasets ("13 DATA"). Where P.F, C.E, N.S, L.C stand for the paralleled formulation, concept enrichment, negative samples and label completion. The latter 3 techniques are realized by the proposed concept dictionary. Numbers in parentheses under LVIS column are $AP_r/AP_c/AP_f$, respectively.

| PRE-TRAINING DATA | P.F | C.E. | N.S | L.C | LVIS | 13 DATA |
|---|---|---|---|---|---|---|
| O365[43] | ✗ | ✗ | ✗ | ✗ | 23.7 (16.6/20.5/27.7) | 30.1 |
| | ✓ | ✗ | ✗ | ✗ | 27.8 (22.2/26.8/29.7) | 30.7 |
| | ✓ | ✓ | ✗ | ✗ | 28.8 (26.0/28.0/30.0) | 33.8 |
| O365[43], GOLDG[23] | ✓ | ✗ | ✗ | ✗ | 28.2 (21.6/25.0/32.2) | 33.7 |
| | ✓ | ✓ | ✗ | ✗ | 32.2 (26.4/30.3/34.9) | 35.9 |
| | ✓ | ✗ | ✓ | ✗ | 30.3 (22.6/27.4/34.2) | 36.3 |
| | ✓ | ✓ | ✓ | ✗ | 34.4 (26.9/33.9/36.3) | 38.8 |
| O365[43], GOLDG[23], YFCC[45] | ✓ | ✓ | ✓ | ✗ | 35.5(32.8/34.9/**36.6**) | 39.9 |
| | ✓ | ✓ | ✓ | ✓ | **35.9**(**33.2/35.7**/36.4) | **43.3** |

Table 4: The impact of the scale of the concept dictionary. The numbers in parentheses indicate the size of the corresponding concept dictionary. The first row means without using the concept dictionary. Zero-shot performances on LVIS [16] are reported.

| MODEL | CONCEPT DICTIONARY | LVIS MINIVAL AP ($AP_r$ / $AP_c$ / $AP_f$) | LVIS VAL AP ($AP_r$ / $AP_c$ / $AP_f$) |
|---|---|---|---|
| DETCLIP-T(B) | / | 28.2 (21.6 / 25.0 / 32.2) | 20.9 (15.3 / 17.5 / 27.1) |
| | O365 (~0.36K) | 27.8 (22.3 / 23.7 / 32.4) | 21.6 (19.3 / 18.7 / 25.8) |
| | O365+THINGS (~1.9K) | 28.1 (20.8 / 24.8 / 32.4) | 20.5 (14.0 / 17.0 / 27.2) |
| | DETECTION + IMAGE-TEXT (~14K) | **34.4 (26.9 / 33.9 / 36.3)** | **27.2 (21.9 / 25.5 / 31.5**) |

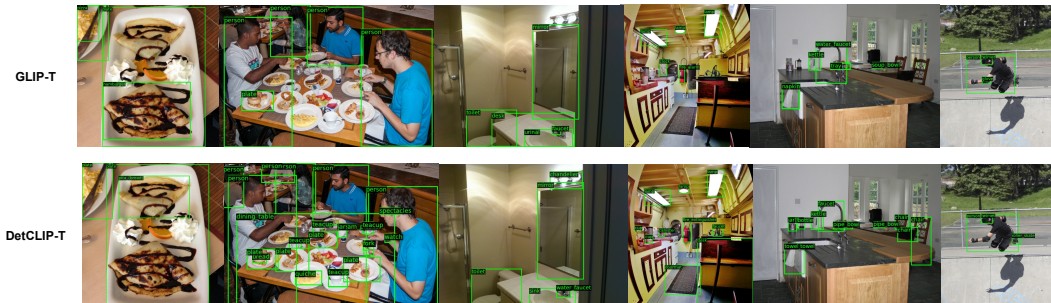

Figure 6: Qualitative prediction results from GLIP-T [28] (first row) and DetCLIP-T (second row) on LVIS [16] dataset. DetCLIP-T can produce more complete and precise predictions than GLIP-T.

the class names to help bridge relationships between different concepts, which boosts the performance to 32.2% (row 5). Sampling negative categories from the concept dictionary also helps better utilize grounding data, improving the performance to 34.4% (row 7). Further introducing image-text pair datasets like YFCC [45] can bring substantial improvement for rare categories (row 8), while utilizing concept dictionary for label completion during pseudo labeling finally improves the overall AP to 35.9% (row 9). A similar performance pattern is also observed on 13 downstream detection datasets.

**Impact of Concept Dictionary's Size.** To study the impact of the scale of the proposed concept dictionary, we build three concept dictionaries with different sizes by using: (1) class names from Objects365 [43]; (2) class names from Objects365 + Things [18]; (3) class names in (2) plus noun phrases extracted from YFCC100m [45]. We equip our DetCLIP-T(B) with these three concept dictionaries and compare their performance in Table 4. It can be seen that using a small size dictionary, e.g., Objects365 + Things, can even bring a performance drop compared to without the dictionary, while scaling up the dictionary with nouns from YFCC can significantly improve the performance.

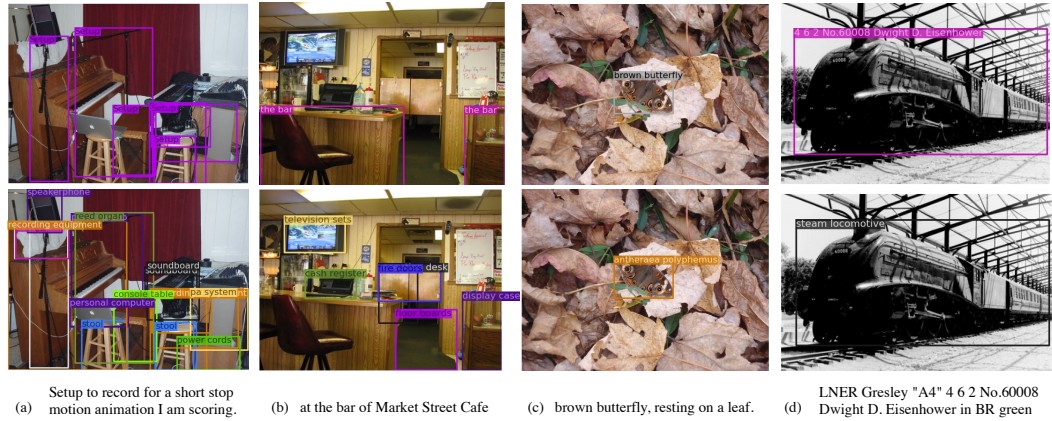

| (a) | Setup to record for a short stop motion animation I am scoring. This is for MFA Thesis project. | (b) at the bar of Market Street Cafe | (c) brown butterfly, resting on a leaf. | (d) | LNER Gresley "A4" 4 6 2 No.60008 Dwight D. Eisenhower in BR green livery at Green Bay Railroad Museum. |

Figure 7: Visualization of pseudo label results on YFCC [45] dataset. The texts below the images are the corresponding captions. Top row: labeling results using the original captions; bottom row: labeling results with the designed concept dictionary. It helps produce higher-quality pseudo labels.

| Box | Text | cat | cat+def$_{cat}$ | cat+def$_{person}$ | person | person+def$_{person}$ | person+def$_{cat}$ |
|---|---|---|---|---|---|---|---|
| (red box) | | 0.30 | 0.33 | 0.28 | ✖ | ✖ | 0.27 |
| (blue box) | | ✖ | ✖ | 0.06 | 0.41 | 0.56 | 0.15 |

Figure 8: Alignment scores when using text inputs with different definition settings. Cross mark means the scores are below the threshold. $def_x$ means we use the definition of category x. Results show that the definition can play an important role in classifying objects.

We speculate this is because a large dictionary can provide rich negative concepts for grounding data, encouraging the model to learn more discriminative features.

**Pseudo labels with concept dictionary.** Fig.7 visualizes the results of label completion via concept dictionary. Multiple effectiveness can be observed. E.g., cases (a) and (b) demonstrate concept dictionary contributes to labeling bbox with the category names not shown in the captions. In (c), a finer-grained pseudo label can be produced, i.e., 'antheraea polyphemus' v.s. the original 'brown butterfly', which helps the learning of rare categories. In (d), label noise in the caption is alleviated.

**Importance of concept enrichment.** To directly illustrate how the category definition helps the model achieve better detection performance, we conduct the experiments to infer with different text inputs with DetCLIP-T utilized. We compare the three different cases, i.e., no definition, true definition and false definition (use definition of another category), and report the alignment score for different objects in Fig.8. Observations are: (1) True definition helps the model to better determine the category (i.e., increases the confidence score); (2) Wrong definition may confuse the model and bring the false positive samples; (3) Both the category name and definition matter to the model.

## 5 Conclusion

In this paper, we propose a novel open-world detection pre-training framework named DetCLIP, aiming at improving the open-domain ability and learning efficiency of the open-world detector. By unifying three kinds of supervision via paralleled concept formulation, our DetCLIP can learn from different domains and enhance the training efficiency. We further propose a concept dictionary module to improve the discovery and coverage of the novel knowledge by importing external knowledge. The proposed usages of the concept dictionary achieves better open-world detection result in terms of both common and rare categories on LVIS. Experiments on multiple downstream detection datasets suggest that our DetCLIP is more powerful than current SOTA open-world detectors such as GLIP [28].

**Acknowledgements** We gratefully acknowledge the support of MindSpore[2], CANN (Compute Architecture for Neural Networks) and Ascend AI Processor used for this research.

---

[2] https://www.mindspore.cn/

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
