# Appendix for DetCLIP: Dictionary-Enriched Visual-Concept Paralleled Pre-training for Open-world Detection

## A    Negative Impacts and Limitations

**Potential Negative Social Impact.** Our method has no ethical risk on dataset usage and privacy violation since all the benchmarks are publicly available and transparent.

**Limitations and Future Works.** The localization ability of the region proposals is still limited by the annotation of the bounding box. More weakly-supervision can be included to learn from image-text pairs. Furthermore, although we prove the effectiveness of our method on the web-collected dataset YFCC [11], we expect to extend our method to larger image-text pair datasets from the Internet.

## B    Dataset Details

In this section, we provide more details of training datasets used in our experiments, which include (1) the approach of generating pseudo detection labels for image-text pair dataset; and (2) a dataset comparison with GLIP [8].

**Pseudo Labeling on Image-Text Pair Data.** We use image-text pair data from the web-collected dataset YFCC100m [11]. To generate pseudo detection labels for image-text pair data, we first use a Region Proposal Network (RPN) pre-trained on Objects365 to extract object proposals. To ensure the quality of proposals, we filter result bounding boxes with objectness scores below a threshold of 0.3 or region area smaller than 6000. This operation also helps significantly reduce the number of proposal candidates and accelerates the pseudo-labeling process.Then a powerful pre-trained CLIP [9] model (ViT-L) is used to predict pseudo class labels for each retained bounding box. To alleviate the partial-label problem, we use concept names from our proposed concept dictionary (Sec.3.2) instead of the raw caption as the text input. Following CLIP, the prompt "a photo of a category." is used to pad a category name into the sentence. Since the proposed dictionary consists of a large number of concepts (i.e., 14k), to accelerate the inference, we pre-compute the text embeddings of all concepts and store them for the later computation. For each proposal bounding box, we first crop it from the raw image, then resize it to $224 \times 224$ and feed it into the visual encoder to obtain the visual embedding. We use the cosine similarity as the classification score, which is computed as

$$s_i^j = f_\theta^I(R_i)g_\phi^T(c_j)^\top \tag{1}$$

where $f_\theta^I$ and $g_\phi^T$ stand for the image and text encoder of the CLIP model; $R_i$ and $c_j$ are $i$-th cropped proposal and $j$-th category, respectively. Both embeddings are L2-normalized before the similarity calculation. After category prediction, a second-stage filtering is adopted to drop proposals with a classification score below $0.24$. Finally, we sample 1M images from the results to form our final training image-text pair data.

**Training Data Comparison (with GLIP [8]).** Table 1 compares the training data used by DetCLIP and GLIP. The explanation of each dataset can be found in the table caption. Our DetCLIP-T uses less than half training data compared to GLIP-T, while DetCLIP-L uses **less than 10%** training data compared to GLIP-L.

Table 1: A training data comparison between DetCLIP and GLIP [8]. Numbers in parentheses indicate the volume of the corresponding dataset. O365-V1 and -V2 are 1st and 2nd version of Objects365 [10] dataset, respectively. 4ODs is a combination of 4 detection datasets, i.e., Objects365 [10], OpenImages [5], Visual Genome [6] and ImageNetBoxes [7]. GoldG is the grounding data introduced by MDETR[4]. Cap4M and Cap24M are web-crawled image-text pair dataset collected by GLIP. YFCC1M is our pseudo-labeled image-text pair data sampled from YFCC100M [11].

|  | Detection | Grounding | Image-text | Total Volume |
|---|---|---|---|---|
| GLIP-T | O365 V1 (0.66M) | GoldG (0.77M) | Cap4M (4M) | ~5.43M |
| GLIP-L | 4ODs (2.66M) | GoldG (0.77M) | Cap24M (24M) | ~27.43 M |
| DetCLIP-T/L | Sampled O365 V2 (0.66M) | GoldG (0.77M) | YFCC1M (1M) | ~2.43M |

Table 2: Manually designed prompts for six downstream detection datasets.

| Dataset | Original Prompt | Manually Designed Prompt |
|---|---|---|
| EgoHands | hand | fist. |
| NorthAmericaMushrooms | CoW
chanterelle | oyster mushroom.
yellow mushroom. |
| Packages | package | a package on the floor. |
| Pothole | pothole | a pit, a sizeable hole. |
| Pistols | pistol | pistol or firearm. |
| ThermalDogsAndPeople | dog
person | dog in heatmap.
person in heatmap. |

## C    More Results on LVIS and 13 Detection Datasets

**More results under GLIP-protocal.** We study the generalization ability of our models by zero-shot transferring them to LVIS [3] full validation set. Following GLIP [8], we use manually designed prompts for some downstream datasets, as illustrated in Table 2. AP for LVIS full validation set is reported in Table 3. Despite using much less training data, DetCLIP models can dominate their GLIP's [8] counterparts in most cases (except $AP_f$ on LVIS for DetCLIP-L). Notably, compared to GLIP, DetCLIP considerably boosts the performance for rare categories, which is an important indicator reflecting models' generalization ability for the open-world detection task. More AP performances for 13 downstream detection datasets can refer to Table 4.

Table 3: Zero-shot transfer performance on LVIS [3] full validation dataset. $AP_r$/$AP_c$/$AP_f$ indicate the AP values for rare, common, frequent categories. 'DH' and 'F' in GLIP [8] baselines stand for the dynamic head [1] and cross-modal fusion.

| MODEL | BACKBONE | PRE-TRAIN DATA | LVIS VAL
AP ($AP_r$ / $AP_c$ / $AP_f$) |
|---|---|---|---|
| GLIP-T(A)[8] | SWIN-T+DH+F | O365 | 12.3 (6.00 / 8.00 / 19.4) |
| GLIP-T[8] | SWIN-T+DH+F | O365,GOLDG,CAP4M | 17.2 (10.1 / 12.5 / 25.2) |
| GLIP-L[8] | SWIN-L+DH+F | 4ODS,O365,GOLDG,CAP24M | 26.9 (17.1 / 23.3 / **36.4**) |
| DETCLIP-T(A) | SWIN-T | O365 | 22.1 (18.4 / 20.1 / 26.0) |
| DETCLIP-T(B) | SWIN-T | O365, GOLDG | 27.2 (21.9 / 25.5 / 31.5) |
| DETCLIP-T | SWIN-T | O365, GOLDG, YFCC1M | 28.4 (25.0 / 27.0 / 31.6) |
| DETCLIP-L | SWIN-L | O365, GOLDG, YFCC1M | **31.2** (**27.6 / 29.6** / 34.5) |

Table 4: Detailed zero-shot transfer AP of DetCLIP on 13 detection datasets [8].

| MODEL | THERMAL | AQUARIUM | RABBITS | MUSHROOMS | AERIALDRONE | PASCALVOC | VEHICLES |
|---|---|---|---|---|---|---|---|
| DETCLIP-T(A) | 34.1 | 12.4 | 70.3 | 44.1 | 10.8 | 53.5 | 52.7 |
| DETCLIP-T(B) | 50.8 | 16.6 | 71.8 | 17.6 | 11.6 | 54.2 | 58.4 |
| DETCLIP-T | 51.3 | 18.5 | 75.2 | **69.2** | 10.9 | 55.0 | 55.7 |
| DETCLIP-L | **53.8** | **25.6** | **75.5** | 67.0 | **20.4** | **56.7** | **64.7** |

| MODEL | EGOHANDS | RACCOON | POTHOLE | PISTOLS | SHELLFISH | PACKAGES | AVG |
|---|---|---|---|---|---|---|---|
| DETCLIP-T(A) | 6.1 | 36.5 | 3.0 | 30.9 | 17.9 | 67.2 | 33.8 |
| DETCLIP-T(B) | 33.8 | 51.9 | 15.2 | 29.2 | 21.8 | 71.5 | 38.8 |
| DETCLIP-T | 34.5 | **54.0** | 14.9 | 31.5 | 24.3 | 68.2 | 43.3 |
| DETCLIP-L | **40.6** | 52.6 | **20.3** | 60.7 | **43.7** | **68.8** | **50.0** |

**More results under VILD-protocal.** To make a more comprehensive evaluation of our method, we also perform experiments under the VILD [2] protocol, i.e., the method is trained on base categories and then evaluated on novel categories using the original LVIS AP metric. We replace the Objects365 part in our training data with LVIS-base, and GoldG and YFCC1M are still included. Including additional data will lead to somehow unfair comparison with VILD but it is necessary since this is the core component in our method to enable zero-shot capability, which differs from VILD that distills knowledge from a pre-trained CLIP model. Note that we implement DetCLIP using the same training/testing setting as in the paper, and do not use techniques such as large-scale jittering and prompt ensemble which is adopted by VILD to boost the performance. The results are shown in the Table 5. Our method (27.3 mAP) outperforms VILD (22.5 mAP) by 4.8% mAP.

Table 5: An comparison with VILD model under VILD protocal.

| MODEL | BACKBONE | LVIS VAL
AP ($AP_r$ / $AP_c$ / $AP_f$) |
|---|---|---|
| VILD [2] | RESNET50 | 22.5 (**16.1** / 20.0 / 28.3) |
| DETCLIP | RESNET50 | **27.3** (14.9 / **25.4** / **34.8**) |

## D   Ablation Studies

**Sequential Formulation with Shuffled Grounding Data (Sec 3.1).** The sequential formulation (e.g., GLIP [8]) is not effective for modeling open-world object detection as a visual-language task since it leads to unnecessary interaction between category names in the attention module. To demonstrate the idea, we randomly shuffle the word order in the grounding training data and report the performance comparison in Table 6. It can be seen that randomly shuffling the word order in the grounding data can even bring a slight improvement  (i.e., +1.4% on LVIS minival) on zero-shot transfer AP for the downstream detection task, indicating that the noun phrase is more critical for the detection task, compared to the context information. Therefore, DetCLIP drops the context information and treats each noun phrase as a paralleled text input, which avoids unnecessary attention among class names and achieves better training efficiency.

Table 6:  Performance comparison of sequential formulation with different grounding data (shuffled word order/original caption). Zero-shot transfer performance on LVIS [3] dataset are reported. The model is trained on Objects365 [10] and GoldG datasets.

| MODEL | GROUNDING DATA | LVIS MINIVAL
AP ($AP_r$ / $AP_c$ / $AP_f$) | LVIS VAL
AP ($AP_r$ / $AP_c$ / $AP_f$) |
|---|---|---|---|
| SEQUENTIAL CONCEPT FORM | ORIGINAL CAPTION
SHUFFLED WORD ORDER | 26.0 (18.0 / 22.8 / 30.3)
**27.4** (**18.6** / **23.8** / **32.3**) | 18.9 (11.7 / 15.8 / 25.6)
**19.9** (**12.5** / **16.4** / **27.0**) |

**Important Role of Class Definition.** In DetCLIP, we augment the class names in the detection dataset with their definitions during both training and inference stage, which is termed as **concept enrichment**. To verify that DetCLIP learns knowledge from class definitions, we compare the performances of including/excluding definitions in text input during the inference stage. Table 7 reports the results. It can be found that adding definitions to class names can significantly improve the zero-shot transfer performance.

**Impact of pre-trained language models in Concept enrichment.**   During training, we use a pre-trained language model to retrieve a definition in our dictionary for concepts without a direct match in WordNet. We conduct experiments to study how the pre-trained language model in the this process affects the final performance. Three different settings are considered: 1. do not use language model, i.e., directly adopt the category name as the input for the concepts not in WordNet; 2. use a pre-trained FILIP text encoder; and 3. use a pre-trained RoBERTa as in GLIP. The results are shown in the Table 8. We can observe that: 1) the concept enrichment procedure can bring significant improvements,  (e.g., +3.6% on rare categories) even without using a pre-trained language model; 2)

Table 7: Effects of concept enrichment during the inference phrase. Text inputs with and without class definition are studied. Zero-shot transfer performance on LVIS [3] dataset is reported. The class definition helps detector better recognize objects.

| Model | Text Input | LVIS minival AP (AP$_r$ / AP$_c$ / AP$_f$) | LVIS val AP (AP$_r$ / AP$_c$ / AP$_f$) |
|---|---|---|---|
| DetCLIP-T(B) | class names | 30.4 (22.4 / 27.1 / 34.8) | 23.2 (14.3 / 20.7 / 29.9) |
| | class names + def. | **34.4 (26.9 / 33.9 / 36.3)** | **27.2 (21.9 / 25.5 / 31.5)** |

using FILIP can further boost the AP performance from 28.3 to 28.8, while using RoBERTa achieves similar performance with no language model is used.

Table 8: Performance comparison of using different pre-trained text encoders in concept enrichment procedure on LVIS minival dataset. The training dataset is Objects365.

| Concepts Enrichment | Pre-trained Text Encoder | LVIS minival |
|---|---|---|
| ✗ | / | 27.8 (22.2/26.8/29.7) |
| ✓ | None | 28.3 (25.8/27.0/29.9) |
| ✓ | RoBERTa-base | 28.2 (24.5/27.3/29.7) |
| ✓ | FILIP text-encoder | **28.8 (26.0/28.0/30.0)** |

**Other Important Training Techniques.** Training a vision-language model that works for the open-world detection task is not easy. We highlight two important training techniques we found in our experiments: (1) using a small learning rate for the pre-trained language backbone, since it helps maintain the language model's knowledge learnt in the large-scale pretraining; and (2) removing the regression loss for non-detection data, since it helps alleviate the negative impact caused by inaccurate localization annotation of grounding/image-text pair data. Table 9 provides the ablation studies of these techniques.

Table 9: Important techniques for training DetCLIP. The learning rate of the image encoder is set to 2.8e-4. Models in this table use sequential formulation as in GLIP [8], since these experiments are conducted during our early-stage exploration.

| Pre-train data | LR (Lang. model) | Reg. Loss | LVIS (MiniVal) |
|---|---|---|---|
| O365 | 2.8e-4 | Det. | 15.9 (7.00/11.3/21.5) |
| O365 | 2.8e-5 | Det. | **23.7 (16.6/20.5/27.7)** |
| O365, GoldG | 2.8e-4 | Det | 22.3 (14.5/17.9/27.6) |
| O365, GoldG | 2.8e-5 | Det. GoldG. | 22.9 (15.3/21.5/25.6) |
| O365, GoldG | 2.8e-5 | Det | **26.0 (18.0/22.8/30.3)** |

# E    Qualitative Results

**More visualizations of pseudo labels with concept dictionary.** Fig. 1 shows extra examples of YFCC data that pseudo labeled with the concept dictionary, as well as their comparisons with the results generated by using the original caption. Concept dictionary alleviates partial-label problem and helps CLIP model provide finer-grained and higher quality pseudo labels.

**Retrieval with Concept Dictionary.** In our **concept enrichment**, to augment a given class name with its definition, we retrieve it in the constructed concept dictionary. If there is an exact match, we directly use the corresponding definition; otherwise, we use semantic similarity computed by a pre-trained language model to find the closest one. Table 10 illustrates some example retrieval results. For class names that are not contained in the dictionary, our method can find proper synonyms.

**Illustrations of Concept Dictionary.** We illustrate some examples in our concept dictionary in Table 11. We observe that the concepts collected from the image-text pair data can cover more fine-grained categories (e.g., cotswold, cuniculus paca) and a wider range of classes (e.g., giant, cathedral).

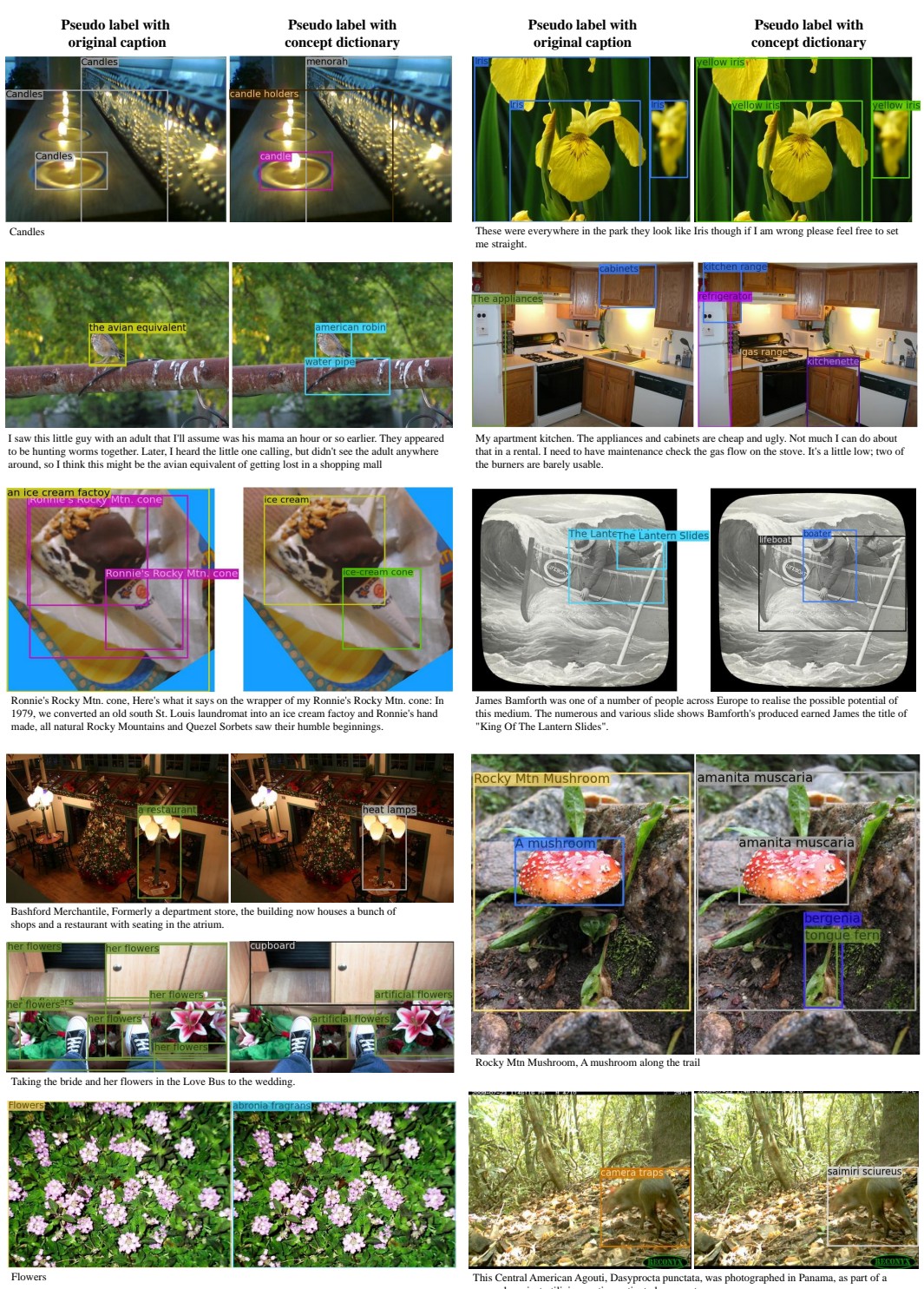

Figure 1: More visualizations of pseudo label results on YFCC [11] dataset. The texts below the images are the corresponding captions. Concept dictionary helps produce higher-quality pseudo labels.

Table 10: Results of retrieval with the proposed concept dictionary. Class names of Objects365 are used as the queries. Our method can retrieve proper synonyms for class names not contained in the dictionary.

| Query Class Name | Retrieved Concept | Definition |
|---|---|---|
| Leather Shoes | Boot | Footwear that covers the whole foot and lower leg. |
| High Heels | Stiletto | A woman's shoe with a thin, high tapering heel. |
| Machinery Vehicle | Truck | An automotive vehicle suitable for hauling. |
| Cosmetics Mirror | Mirror | Polished surface that forms images by reflecting light. |
| Induction Cooker | Hotplate | A portable electric appliance for heating or cooking or keeping food warm. |
| Hoverboard | Rollerblade | Trademark an in line skate. |

Table 11: Examples of our concept dictionary. The upper part of the table shows the concepts collected from the detection datasets, while the lower part shows the concepts collected from the image-text pair dataset, i.e., YFCC100M [11].

| Concept | Definition |
|---|---|
| Collected from Large-scale Detection Datasets. | |
| Cup | A small open container usually used for drinking; usually has a handle. |
| Chips | strips of potato fried in deep fat. |
| Chair | A seat for one person, with a support for the back. |
| Eraser | an implement used to erase something. |
| Gloves | The handwear used by fielders in playing baseball. |
| Recorder | equipment for making records. |
| Street Lights | A lamp supported on a lamppost; for illuminating a street. |
| Collected from Image-text Pair Data. | |
| Pet | A domesticated animal kept for companionship or amusement. |
| Pod | The vessel that contains the seeds of a plant (not the seeds themselves). |
| Taro | Edible starchy tuberous root of taro plants. |
| Shrub | A low woody perennial plant usually having several major stems. |
| Salmon | Any of various large food and game fishes of northern waters. |
| Brewery | A plant where beer is brewed by fermentation. |
| Pottery | Ceramic ware made from clay and baked in a kiln. |
| Giant | Any creature of exceptional size.giant and any creature of exceptional size. |
| Pagoda | An Asian temple; usually a pyramidal tower with an upward curving roof. |
| Fresco | A mural done with watercolors on wet plaster. |
| Wildflower | Wild or uncultivated flowering plant. |
| Water tower | A large reservoir for water. |
| Basin | A bowl-shaped vessel; usually used for holding food or liquids. |
| Cotswold | Sheep with long wool originating in the Cotswold Hills. |
| Insect | Small air-breathing arthropod. |
| Booth | A table (in a restaurant or bar) surrounded by two high-backed benches. |
| Office | Place of business where professional or clerical duties are performed. |
| Cab | A compartment at the front of a motor vehicle or locomotive where driver sits. |
| Gable | The vertical triangular wall between the sloping ends of gable roof. |
| Hotel | A building where travelers can pay for lodging and meals and other services. |
| Cathedral | Any large and important church. |
| Restaurant | A building where people go to eat. |
| Library | A room where books are kept. |
| Courtyard | An area wholly or partly surrounded by walls or buildings. |
| Footbridge | A bridge designed for pedestrians. |
| Cuniculus paca | Large burrowing rodent of South America and Central America. |