# OpenReview forum: "DetCLIP: Dictionary-Enriched Visual-Concept Paralleled Pre-training for Open-world Detection"
_NeurIPS.cc/2022/Conference — NeurIPS 2022 Accept_

### Official Review · Reviewer_aMQW · 2022-07-09

**Rating:** 6
**Confidence:** 5
**Soundness:** 3 good
**Presentation:** 3 good
**Contribution:** 2 fair

**Summary:**

This paper proposes an extension of GLIP for Open World Detection. The extension comprises two parts: one methodological by introducing a so-called parallel processing of the text data, and one more data related that allows the unification of many heterogeneous datasets for training. The latter also proposes a richer description of the categories -- deemed concepts in the paper -- through wordnet.
The paper reports significant improvements of GLIP. It is mainly evaluated on LVIS dataset

**Questions:**

On the negative side:
- the methodological contribution of the parallel processing is rather thin and is actually the first thing that would come to mind when reading the original GLIP paper.
- most of the improvements and contributions come from the dataset side. Although the framework for creating the dataset is worthy, it is more an engineering approach rather than an actual methodological contribution. After reading the paper you get mostly the impression that the authors created a much better dataset and then trained a (straightforward) extension of GLIP on it.
- The evaluation on LVIS is sound and convincing. However with such big and heterogeneous data used, the zero shot capabilities of the method are not clear. Can the authors also evaluate following the VILD protocol (https://arxiv.org/pdf/2104.13921.pdf) where the method is trained on base classes and then explicitly evaluated on unseen classes?

**Final Recommendation**
As said I will stick to my original score which is line with the score from other Rs.

**Limitations:**

I didn't see a paragraph for this in the paper.

**Strengths And Weaknesses:**

On the positive side:
- the authors study an important problem in literature that of open-world object detection where not many solutions are available
- the parallel extension to GLIP, although straightforward, makes sense. The original GLIP sequentual formulation was problematic and did not scale well with a large number of categories
- the authors did a good job in terms of proposing a framework for unifying many established and heterogeneous datasets including standard detection dataset, grounding data, and image/caption pairs. Actually, in my opinion, this is the biggest contribution of the paper.
- the improvements over GLIP are significant. Even when trained on the same data, +5.1% is reported (table 3).

---

> ### Author Response · Authors · 2022-08-02
> **To Reviewer aMQW**
>
> We thank the reviewer for the positive feedback and insightful comments! All comments are summarized and addressed as follows.
>
> **R4Q1. The parallel processing is rather thin and is actually the first thing that would come to mind.**
>
> We argue that the proposed paralleled formulation for unifying heterogeneous datasets is new. We also demonstrate this idea is neat yet effective. Specifically, it achieves significant performance improvement (+4.1% mAP on LVIS) compared with the sequential formulation used in GLIP, as compared in row1 and row2 in the Tabel 3 of the main paper. And paralleled formulation is also efficient for both training (5x faster than GLIP) and inference (19x faster than GLIP), as demonstrated in Table 2 of the main paper. These explorations can serve as a valuable guideline for the community to design vision-language models for open-domain detection tasks.
>
> **R4Q2. Most of the improvements and contributions come from the dataset side. It is more an engineering approach rather than an actual methodological contribution.**
>
> It should be noted that our main focus is not simply using more training data for learning, but to investigate and design an effective framework able to incorporate large-scale heterogeneous data sources to largely boost the learning of wider range of visual-language knowledge, and to provide a broader-generalization model for open-domain detection, which is a critically important direction that also shares the same motivation and focus as GLIP. While directly comparing to the pioneering work GLIP in this direction, our proposal has several key technical advantages and contributions: (i) we propose a neat yet effective paralleled formulation, (ii) we introduce a novel concept enrichment method with the proposed concept dictionary to explicitly build the relationship between categories (iii) we explore label completion technique to better utilize large-scale image-text pair data. Moreover, as demonstrated in Q1, our method can significantly improve the performance as well as the efficiency compared to GLIP.
>
> **R4Q3. Can the authors evaluate the proposed method following the VILD protocol?**
>
> GLIP protocol and VILD protocol are both common practices to evaluate the zero-shot detection performance. The former is a stronger and more challenging setting since it does not make any prior assumptions about the data distribution of downstream tasks, For instance, GLIP and DetCLIP can directly perform the evaluation on 13 detection datasets with highly varied distributions, while the VILD protocol still assumes the testing images and training images should come from the same dataset (LVIS). Besides, achieving a good performance under the GLIP protocol is more difficult and that is why we should use large-scale and heterogeneous training data.
>
> To make a more comprehensive evaluation of our method, we also perform experiments under the VILD protocol as suggested. We replace the Objects365 part in our training data with LVIS-base, and GoldG and YFCC1M are still included. Including additional data will lead to somehow unfair comparison with VILD but it is necessary since this is the core component in our method to enable zero-shot capability, which differs from VILD that distills knowledge from a pre-trained CLIP model. The result is shown in the following Table. Our method (27.3% mAP) outperforms VILD (22.5% mAP) by 4.8% mAP. Note that due to the limited time of the rebuttal, we implement DetCLIP using the same training/testing setting as in the paper, and do not use techniques such as large-scale jittering and prompt ensemble which is adopted by VILD to boost the performance. The comparison is updated in the revised paper (Table 5 in Appendix C).
>
> |  Model  | Backbone | LVIS val AP |
> | :-----: | :------: | :---------: |
> |  VILD   | ResNet50 |    22.5     |
> | DetCLIP | ResNet50 |  **27.3**   |
>
> **R4Q4. No limitation paragraph.**
>
> Please refer to the Appendix A.

---

> > ### Comment · Reviewer_aMQW · 2022-08-04
> > **keep my original score**
> >
> > I appreciate the authors' effort to address my comments, especially evaluating their method using the VILD protocol. This result should be included in the final version (at least in the supplementary). I still believe I was right in my original assessment about the advantages and disadvantages of the proposed method. Overall, I still believe the paper makes a good contribution and hence I stick to my original score.

---

### Official Review · Reviewer_rAnj · 2022-07-10

**Rating:** 6
**Confidence:** 5
**Soundness:** 2 fair
**Presentation:** 3 good
**Contribution:** 2 fair

**Summary:**

The paper is developed based on GLIP:

- To solve the problems of inefficient interaction between categories and the restriction of max input sentence length, it proposes a parallel method. Compared to the sequential input in GLIP, the paper extracts phrases as the input in a parallel manner.
- To build implicit relationships between input phrases, it proposes a concept dictionary to provide the prior knowledge.

The paper is verified in the task of zero-shot detection on LVIS.


**Questions:**

- The notations in Formula (1) are odd due to the upper letter T and Transpose. Please consider changing it.
- Experiments mainly use Swin-Transformer as the backbone. Can the backbone be ViT?

**Limitations:**

None.

**Strengths And Weaknesses:**

Strengths:
- The framework does make sense. I agree with the point that extracting the noun phrase is efficient for alignment between regions and concept words.
- The proposed large-scale concept dictionary could be a future valuable toolkit for future open-vocabulary or vision-language research if it can be open-released.
- Compared to GLIP, improvements are large.

---

> ### Author Response · Authors · 2022-08-02
> **To Reviewer rAnj**
>
> We thank the reviewer for the positive feedback and insightful comments! All comments are summarized and addressed as follows.
>
> **R3Q1. Experiments mainly use Swin-Transformer as the backbone. Can the backbone be ViT?**
>
> Our method is not limited to a specific backbone. Other vision backbones like ViT or ResNet can be considered as a choice. Swin-Transformer is more suitable for object detection tasks compared to ViT since the window-attention mechanism can help significantly reduce the computational cost when the input has a high resolution. Another consideration of using Swin-Transformer is it can lead to a fair comparison with GLIP if we use the same backbone. If someone would like to use ViT as the backbone, we recommend following the method in [1].
>
> [1] Li, Yanghao, et al. "Exploring plain vision transformer backbones for object detection." arXiv preprint arXiv:2203.16527 (2022).
>
> **R3Q2. The notations in Formula (1) should be changed.**
>
> Thanks for your suggestion, we will change it in the revision.

---

### Official Review · Reviewer_82yp · 2022-07-11

**Rating:** 6
**Confidence:** 5
**Soundness:** 3 good
**Presentation:** 3 good
**Contribution:** 3 good

**Summary:**

This paper explores a parallel visual-concept pretraining method for open vocabulary object detection by resorting to knowledge enrichment from a designed concept dictionary.
They make the following contributions:
1.a novel paralleled concept formulation to improve learning efficiency.
2.a novel concept dictionary to enrich the prompt text concepts during joint pretraining.
3.DetCLIP outperforms the state-of-the-art GLIP.

**Questions:**

See above

**Limitations:**

See above

**Strengths And Weaknesses:**

Strengths
1.  The method is simple and effective and I like the examples author offered here and there.
2.  The model significantly improves the learning efficiency.
3.  The experiment results seem solid. They evaluate the method on LVIS and transfer the model to 13 downstream detection datasets.

Weaknesses
I think the writing of the abstract and introduction needs to be significantly improved: they are too long especially for the abstract. And also many concepts were used directly without explanation/definitions/references at the first hand or not at all. For instance, what is GLIP, DetCLIP-T, GLIP-T, positive samples v.s. negative samples ? What is the grounding, what's its difference to image-text pair data ? what's the partial label problem ?

Authors claimed their method is much more efficient and effective than GLIP in terms of data usage. However I notice that they have also utilized other models such as CLIP , FILIP. Are these models considered as part of the proposed method (are they multi-stage) ? How would this compare to GLIP ?

L157 Can you provide further explanation that why interaction between categories is unnecessary?

L169: Why context information is just removed. I know the context cannot fed into the text encoder individually. Maybe it can be added to every individual concept phrase. Any study for this  ?

L243 : What is partial positive concept ?  Is it derived from the partial labelling problem, I don't see a clue.

Last, bold text is normally for title/subtitle as a standard practice. Be using it properly!

---

> ### Author Response · Authors · 2022-08-02
> **To Reviewer 82yp**
>
> We thank the reviewer for the positive feedback and insightful comments! All comments are summarized and addressed as follows.
>
> **R2Q1. Authors claimed their method is much more efficient and effective than GLIP in terms of data usage. However, I notice that they have also utilized other models such as CLIP, FILIP. Are these models considered as part of the proposed method (are they multi-stage)? How would this compare to GLIP?**
>
> Yes, these models are considered as part of the proposed method and they are multi-stage.
> Although our method involves external models such as CLIP and FILIP, the comparison with GLIP on data efficiency is fair and clear. Specifically, we can compare the components in our method where the external models are introduced:
>
> 1) We use weights of a vision-language pre-trained FILIP text-encoder as the initialization of our text-encoder, while GLIP loads weights from a language pre-trained RoBERTa. Both models are transformer models but FILIP has lower complexity, i.e., 63.3 M parameters of FILIP-base v.s. 124.8 M parameters of RoBERTa-base, which should not lead to an advantage of our model in comparison. Indeed, in our early stage of experiments, we have also tried using RoBERTa-base as the text encoder, and it achieves comparable results compared to using FILIP's text encoder.
>
> 2) In our concept enrichment technique, we use a pre-trained FILIP text-encoder to retrieve definitions for concepts without a direct match in WordNet. In Q2 of Reviewer 5KVn, we show that this process is robust to the choice of language model and it can achieve a strong performance even without using a language model.
>
> 3) We use CLIP to perform pseudo labeling for image-text pair data while GLIP uses a GLIP-L model trained on detection/grounding data. The effects of using different models during this process cannot be directly compared since we follow different training stages, i.e., we use 2 stages of pseudo-labeling->training while GLIP follows 3 stages of training->pseudo-labeling->re-training. However, to show our method can achieve a better data usage efficiency, we can exclude the pseudo labeling process for both methods and compare the results trained with only detection+grounding data. Under this setting, as shown in the updated Table 1 in the main paper, our DetCLIP-T(B) (34.4% mAP) can outperform the counterpart GLIP-C (24.9% mAP) by 9.5% mAP on LVIS.
>
> **R2Q2. Further explanation for why interaction between categories is unnecessary.**
>
> The interaction between categories can cause inefficient learning and an inconsistency between the training and testing phases. Specifically, in our method, the text embedding of the category name serves as the classification weight of the detector. In the sequential formulation, due to the interaction between category names, the text embedding of each category will depend on other categories which are input together with it, and the order in which they are arranged. This will lead to dynamic classification weights for each category. In other words, the classification weight of a category depends on all the category names and also depends on how we organize them. That is to say, we expect the model to learn visual embeddings that can work on dynamic classification weights, which increases the learning difficulty and hurts the final performance. As demonstrated in our experiments (Fig.4 and Table 3 in the main paper), replacing the sequential formulation with paralleled formulation can significantly improve learning efficiency as well as performance.
>
> On the other hand, to make sequential formulation work, GLIP samples a large number of different category names at each training iteration and shuffles their order as the input to the text-encoder. The purpose of these practices is to reduce the influence of contextual information in the text on each class name, which is contrary to the purpose of interaction.
>
> **R2Q3. Why context information is just removed? Maybe it can be added to every individual concept phrase.**
>
> We remove the context information since our paper targets at the open-world detection which concentrates on recognizing the class of each object. Specifically, in our setting, we keep the adjective for each concept and remove the conjunction and verb in each caption in grounding datasets, which follows the way the grounding datasets annotate objects.
>
> We conduct the experiments (L150-153 in the revised main paper \& Table 6 in the Appendix D) to show that randomly shuffling the word order in the grounding training data for GLIP can even bring a slight improvement for the downstream detection task, indicating that the noun phrase is more critical for the detection task, compared to the context information.
>
> We agree including more context information (e.g., the verb) into concept phrases may be helpful for the model to better distinguish more fine-grained patterns, e.g., actions of objects, which is not considered in this paper.

---

> > ### Author Response · Authors · 2022-08-02
> > **To Reviewer 82yp (cont.)**
> >
> > **R2Q4. What is partial positive concept and is it derived from the partial labelling problem?**
> >
> > Yes, it is derived from the partial labelling problem. Specifically, the "lack of annotations of partial positive concepts" indicates that some concepts in images are not annotated with captions in grounding and image-text datasets. We will make it more clear in revision.
> >
> > **R2Q5. Writing issues.**
> >
> > Thanks for you suggestions. We will refine the abstract/introduction to make them more clear and neat, and modify the bold text in the revision.
> >
> > **R2Q6. Many concepts were used directly without explanation/definitions/references at the first hand.**
> >
> > Thanks for your kindly reminder, we will update this information in the revision. The explanations are listed as follows:
> >
> > 1) The GLIP denotes the method in [1] and the DetCLIP is our proposed method.
> >
> > 2) `-T' denotes the model that adopts the Swin-T transformer as the image encoder.
> >
> > 3) The positive/negative samples for an image denote the categories (or concepts) labeled/not labeled in the ground truth of the image, respectively. In our paralleled formulation, for each training image, we use ground truth categories/concepts as positives and sample negatives from a pool of candidates to perform classification learning. The pool of negative candidates for detection data is the set of categories not appear in the ground truth, and for grounding/image-text pair data it is our proposed concept dictionary.
> >
> > 4) The grounding data denotes the data annotated with fine-grained correspondence between
> > text phrases in a sentence and objects in an image.
> >
> > 5) Partial label problem stands in some grounding or image-text pair datasets, only the main objects that people care about are labeled in the caption.
> >
> > [1] Li, Liunian Harold, et al. "Grounded language-image pre-training." Proceedings of the IEEE/CVF Conference on Computer Vision and Pattern Recognition. 2022.

---

### Official Review · Reviewer_5KVn · 2022-07-12

**Rating:** 6
**Confidence:** 4
**Soundness:** 4 excellent
**Presentation:** 3 good
**Contribution:** 3 good

**Summary:**

The work proposes to build a concept dictionary which enables to perform visual-concept pretraining in a unified way across multiple heterogeneous datasets for open-world detection. With the proposed dictionary design, the proposed method can even perform partial annotation enrichment for label completion and negative concept sampling to further improve the performance.

**Questions:**

The experimental results are very thorough covering different open-vocabulary datasets, and also show promising performance improvement. I have few questions.

1. Is the definition from the wordNet necessary to be used? If we only use the unified label, how much will it affect the performance?
The following work is not targeted at open-vocabulary detection, but it also proposes to perform contrastive learning using the unified space.

Yang, Jianwei, Chunyuan Li, Pengchuan Zhang, Bin Xiao, Ce Liu, Lu Yuan, and Jianfeng Gao. "Unified contrastive learning in image-text-label space." In Proceedings of the IEEE/CVF Conference on Computer Vision and Pattern Recognition, pp. 19163-19173. 2022.


2. How is the pseudo labeling for label completion of using the all the concepts in the dictionary conducted? The dictionary size is usually very large (e.g., 14k). The paper describes to add the concepts in the dictionary as the addition category inputs.
3. For concept enrichment and partial concept enrichment, how would the pretrained encoder FILIP affect the final performance? Do the authors try other text-encoders?
4. How is the proposed method compared with recently released work, GLIPv2 (optional)?
https://arxiv.org/abs/2206.05836

**Limitations:**

The authors do address the limitations and why there is no potential negative societal impact of their work.

**Strengths And Weaknesses:**

Strength:

1. The built concept dictionary can be used to convert the annotations of heterogeneous datasets, including detection, visual grounding, etc, into a unified one.
2. With the concept dictionary, the proposed method can perform concept enrichment with the assistance of other pretrained text encoder, FILIP in addition to negative concept sampling and partial annotation enrichment for those labels which are not well annotated in the existing datasets.
3. The performance improvement in LVIS for rare and frequent categories is significant as compared with other previous state-of-the-art method.

Weakness:

1. The construction of the concept dictionaries rely on the wordNet, and thus  the concepts are mainly composed of those which are well-defined in WordNet.
2. For the concept enrichment, it still requires a pre-trained text encoder, like FILIP, to match related concepts. The quality will depends on the reliability of the pretrained text-encoder used.
3. When resolving the name ambiguity among different datasets, only the text part is used without exploiting the corresponding image information.

---

> ### Author Response · Authors · 2022-08-02
> **To Reviewer 5KVn**
>
> We thank the reviewer for the positive feedback and insightful comments! All comments are summarized and addressed as follows.
>
> **R1Q1. The construction of the concept dictionaries is constrained by the WordNet.**
>
> Yes, the current concepts dictionary relies on the concepts in WordNet since it is large enough to cover most object concepts in our daily life.
>
> Specifically, according to [1], WordNet contains approximately 57,000 noun word forms organized into approximately 48,800 word meanings (synsets), which can cover more than 95\% concepts of the currently available detection datasets like LVIS and Object365.
>
> While making dictionaries larger and more diverse promises to further improve the performance, one of our main purposes is to demonstrate the effectiveness of the idea of the concept dictionary. As shown in Table 3 of the main paper, our models using the current dictionary improves the LVIS rare category performance from 22.2 to 26.0 when trained with Objects365 and from 21.6 to 26.4 when trained with Objects365+GoldG, respectively. We will consider enriching our dictionary by including concepts from broader sources like Wikipedia in the future.
>
> [1] Miller, George A., et al. "Introduction to WordNet: An on-line lexical database." International journal of lexicography 3.4 (1990): 235-244.
>
> **R1Q2. The concept enrichment relies on a pre-trained language model, how would it affect the final performance? Do the authors try other text-encoders?**
>
> During training, we use a pre-trained language model to retrieve a definition in our dictionary for concepts without a direct match in WordNet. Due to the wide coverage of WordNet, only a small fraction (less than 10%) of concepts require retrieving their definitions. Therefore we can expect that the choice of the language model would lead to limited influence on the performance.
>
> We also conduct additional experiments to study how the pre-trained language model in the concept enrichment affects the final performance. Three different settings are considered: 1) do not use the language model, i.e., directly adopt the category name as the input for the concepts not in WordNet;  2) use a pre-trained FILIP text encoder; 3) use a pre-trained RoBERTa as in GLIP. The results are shown in the following table.  We can observe that: 1) the concept enrichment procedure can bring significant improvements, (e.g., +3.6% on rare categories) even without using a pre-trained language model; 2) using FILIP can further boost the AP performance from 28.3 to 28.8, while using RoBERTa achieves similar performance with no language model is used. We update these experimental results in the revised paper (refer to the Paragraph 'Impact of pre-trained language models in Concept enrichment.' and Table 9 in Appendix D).
>
> Table 1:  Performance comparison of using different pre-trained text encoders in concept enrichment
> procedure on LVIS minival dataset. The training dataset is Objects365.
>
> | Concept Enrichment | Pre-trained Text Encoder |  LVIS minival AP (r/c/f)  |
> | :----------------: | :----------------------: | :-----------------------: |
> |         X          |            /             |   27.8 (22.2/26.8/29.7)   |
> |      &#10004;      |           None           |   28.3 (25.8/27.0/29.9)   |
> |      &#10004;      |       RoBERTa-base       |   28.2 (24.5/27.3/29.7)   |
> |      &#10004;      |    FILIP text-encoder    | **28.8 (26.0/28.0/30.0)** |
>
> **R1Q3. When resolving the name ambiguity among different datasets, only the text part is used without exploiting the corresponding image information.**
>
> Thanks for your suggestion, images can provide valuable information for resolving the name ambiguity, e.g., "mouse" can be an animal or equipment. We will consider this direction in our future work.
>
> **R1Q4. Is the definition from WordNet necessary to be used? How would it compare to using the unified label?**
>
> Yes, the definition from the WordNet is important for the performance since it can explicitly provide the useful relationships between various concepts. We show the performance of only using the unified label (similar to the listed CVPR22 paper [2]) in row4 of Table 3 in the main paper, where O365 and GoldG are used for training and no concept enrichment is adopted. Comparing it to the results of row5, where the concept enrichment is used, we can find that augmenting category names with definitions achieves significant improvements (+4% on LVIS and +3% on 13 detection datasets)
>
> [2] Yang, Jianwei, Chunyuan Li, Pengchuan Zhang, Bin Xiao, Ce Liu, Lu Yuan, and Jianfeng Gao. "Unified contrastive learning in image-text-label space." In Proceedings of the IEEE/CVF Conference on Computer Vision and Pattern Recognition, pp. 19163-19173. 2022.

---

> > ### Author Response · Authors · 2022-08-02
> > **To Reviewer 5KVn (cont.)**
> >
> > **R1Q5. How the pseudo labeling for label completion is conducted since dictionary size is usually very large (14K)?**
> >
> > During pseudo labeling, since we use a text encoder that has no interaction with the vision encoder, the text embedding of all concepts can be pre-computed and stored for the later computation. This process requires all concepts to forward the text encoder only once, which is quite efficient.
> > Then the text embeddings are used as the classification weights for object proposals, where the category number is the number of concepts, i.e., 14k. Before calculating the classification scores, we first use a series of conditions to filter proposal candidates with low quality (as described in the L16-18 in Appendix B), which significantly reduces the number of object candidates. By doing this, we can directly calculate the similarities between object proposals and concept texts without encountering memory issues. If a larger scale of concepts is involved, we can split the concepts into multiple chunks and compute the similarities for each chunk separately. We have updated more implementation details about this part in the Paragraph **Pseudo Labeling on Image-Text Pair Data** in Appendix B.
> >
> > **R1Q6. The comparison with GLIPv2.**
> >
> > The DetCLIP focuses on building an effective paralleled framework with external knowledge for open-world object detection while GLIPv2 is proposed to serve different vision tasks, which is orthogonal to our work.
> >
> > For zero-shot detection performance comparison, our DetCLIP-T still outperforms GLIPv2-T (refer to Table 2 of GLIPv2 paper) by a large margin (35.9% v.s. 29.0%) on LVIS minival. We have updated this comparison in Table 1 (row 9) of the revised paper. Note that the performance of the other two models GLIPv2-B and GLIPv2-H cannot be directly compared with our DetCLIP since they include images of LVIS during the training, which is thus not zero-shot.

---

### Author Response · Authors · 2022-08-02
**General Response to All Reviewers**

We thank all the reviewers for their time, insightful suggestions, and valuable comments. We are glad that **ALL** reviewers give positive feedback and find our method simple and effective, an elegant concept parallel framework that scales well with a large number of categories (reviewer 'rAnj' and 'aMQW'), with significant improvements over GLIP on LVIS and 13 downstream detection datasets (reviewer '5KVn', '82yp' and 'aMQW').

We respond to each reviewer's comments in detail below. We have also revised the main paper and appendix according to the reviewers' suggestions. The main changes are listed as follows:

- In Table 1, we add comparisons with GLIP-C and GLIPv2-T.

- In Appendix B, we elaborate on more implementation details about how to conduct pseudo labeling for YFCC dataset when using a large-size concept dictionary.

- In Appendix C, we add an experiment to evaluate our method under VILD protocol.

- In Appendix D, we add experimental results about the impact of the pre-trained language model for concept enrichment.

- As suggested by Reviewer2, we revise the abstract to make it more clear and neat and modify the improper bold text.

Note that we marked the revisions in blue. We hope that our efforts address the reviewers' concerns.

---

### Meta-Review · Area_Chair_W8nq · 2022-08-27

**Recommendation:** Accept
**Confidence:** Certain

**Metareview:**

The paper receives overall positive reviews and rebuttal has resolved the reviewer's concerns. Reviewers agree that the paper proposes a simple yet effective approach to enrich language concepts to learn better region-concept alignment for object detection. The approach is supported by solid empirical evidence on the LVIS dataset and 13 DATA. AC agrees the methodology of processing data is worth sharing to a broader audience and recommends to accept the paper.

**Award:**

No

---

### Decision · Program_Chairs · 2022-09-14

Accept